Extended Abstract Track

# Geometry reveals an instructive role of retinal waves as biologically plausible pre-training signals

**Andrew Ligeralde**                                      LIGERALDE@BERKELEY.EDU
*University of California, Berkeley*
**Miah N. Pitcher**                                       MIAH_PITCHER@BERKELEY.EDU
*University of California, Berkeley*
**Marla B. Feller**                                       MFELLER@BERKELEY.EDU
*University of California, Berkeley*
**SueYeon Chung**                                         SCHUNG@FLATIRONINSTITUTE.ORG
*Flatiron Institute*

**Editors:** Sophia Sanborn, Christian Shewmake, Simone Azeglio, Arianna Di Bernardo, Nina Miolane

## Abstract

Prior to the onset of vision, neurons in the developing mammalian retina spontaneously fire in correlated activity patterns known as retinal waves. Experimental evidence suggests retinal waves strongly influence sensory representations before the visual experience. We aim to elucidate the computational role of retinal waves by using them as pre-training signals for neural networks. We consider simulated activity patterns generated by a model retina as well as real activity patterns observed experimentally in a developing mouse retina. We show that pre-training a classifier with a biologically plausible Hebbian learning rule on both simulated and real wave patterns improves the separability of the network's internal representations. In particular, the pre-trained networks achieve higher classification accuracy and exhibit internal representations with higher manifold capacity when compared to networks with randomly shuffled synaptic weights, particularly for noisy data.

**Keywords:** retinal waves, development, vision, representation, geometry, manifold, classification, biologically plausible, pre-training

## 1. Introduction

How the brain learns to solve visual object recognition is still unknown. Evidence suggests that sensory representations of distinct objects in the early visual system are tangled together and gradually untangle as they are transformed and re-mapped in a feedforward manner along the ventral stream (DiCarlo et al., 2012). However, we do not know the precise mechanism by which the brain learns the appropriate sequence of mappings to sufficiently separate objects for recognition in higher visual areas.

Interestingly, many key aspects of visual function are well-established before the visual experience, such as topographic maps, orientation selectivity, and ocular dominance (Espinosa and Stryker, 2012), suggesting external stimuli are not necessary for the initial development of the visual system, and axon targeting can largely be learned by internally generated signals such as spontaneous neural activity and molecular guidance cues. Both have been implicated in establishing retinal axonal projections, though their relative importance is unknown (Feller and Kerschensteiner, 2020). Here, we focus on spontaneous

# Extended Abstract Track

activity in the form of retinal waves, correlated patterns of propagating, wave-like activity that occur in the absence of light responses among groups of retinal ganglion cells (RGCs) early in mammalian development (Arroyo and Feller, 2016). Experimental and computational evidence suggests that retinal waves instruct the formation of retinotopic maps, enabling RGC axons to reach their targets in the superior colliculus and lateral geniculate nucleus before the eyes even open (Cang et al., 2005; Chandrasekaran et al., 2005; Huberman et al., 2006; Markowitz et al., 2012; Hunt et al., 2012). Given that object recognition in the visual system presumably depends on the representations induced by these axonal projections and that these projections are well formed prior to the visual experience, here we explore whether retinal waves are sufficient for learning the mappings that enable core object recognition.

Our computational work predicts that the spatiotemporal properties of retinal waves can instruct synaptic projections to higher visual areas such that object representations become more linearly separable. Using a simple feedforward network trained to classify MNIST digits (Raghavan and Thomson, 2019), we demonstrate that training the input layer weights on retinal wave patterns with a biologically plausible Hebbian learning rule prior to training the classifier layer on MNIST improves the linear separability of the network's internal representations as well as the classifier accuracy. To do this, we turn to the framework of manifold geometry, in which the collective response of neurons in the network to different presentations of a particular object comprises an "object manifold" (Chung and Abbott, 2021). In particular, we examine three metrics that describe the object manifold geometry — capacity, dimension, and radius (Chung et al., 2018) — and show that the object manifolds in the networks pre-trained with both simulated and experimentally obtained retinal wave patterns are more easily separable than those in control networks whose weights are random permutations of the pre-trained network weights or random projections. We also show that pre-trained networks are more robust and more effectively maintain geometrically favorable internal representations when trained and tested on noisy versions of the data. This suggests that the spatial structure learned from retinal waves (in the absence of the task data) is relevant for visual object recognition and implies an instructive role for retinal waves prior to the visual experience.

## 2. Results

We consider feedforward networks consisting of an input layer, a layer with pre-trained weights (the receptive fields) and ReLU activations, and a support vector machine (SVM) as the classifier/output layer (Appendix B). Only the SVM layer is trained on the MNIST data, such that it learns to classify the network's internal representations of the digits. We present results for five networks: "Pre-trained (sim.)", a network with receptive fields pre-trained on simulated retinal waves according to the dynamics in (Raghavan and Thomson, 2019), "Pre-trained (real)"; a network with receptive fields pre-trained on real retinal wave data (Appendix A); "Scrambled (sim.)" and "Scrambled (real)", networks in which each receptive field is the result of randomly permuting the pixels of its corresponding receptive field in "Pre-trained (sim.)" and "Pre-trained (real)", respectively; and a network whose hidden layer is a Gaussian random projection "Control (rand. proj.)", which reduces dimensionality while preserving geometrical properties of the input (Bingham and Mannila,

Extended Abstract Track

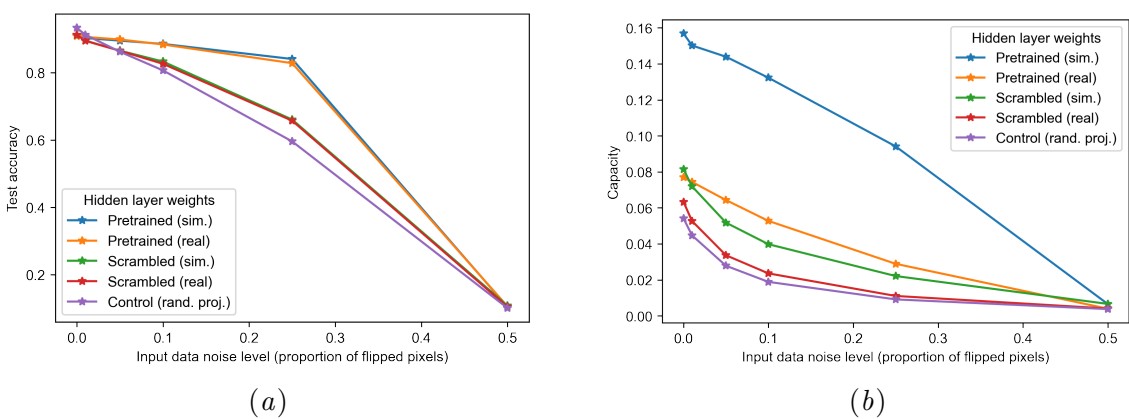

(a)                                       (b)

Figure 1: (a) Accuracy of networks on MNIST with increasing noise level (proportion of flipped pixels). (b) Capacity of network hidden layer representations averaged across MNIST digit class manifolds.

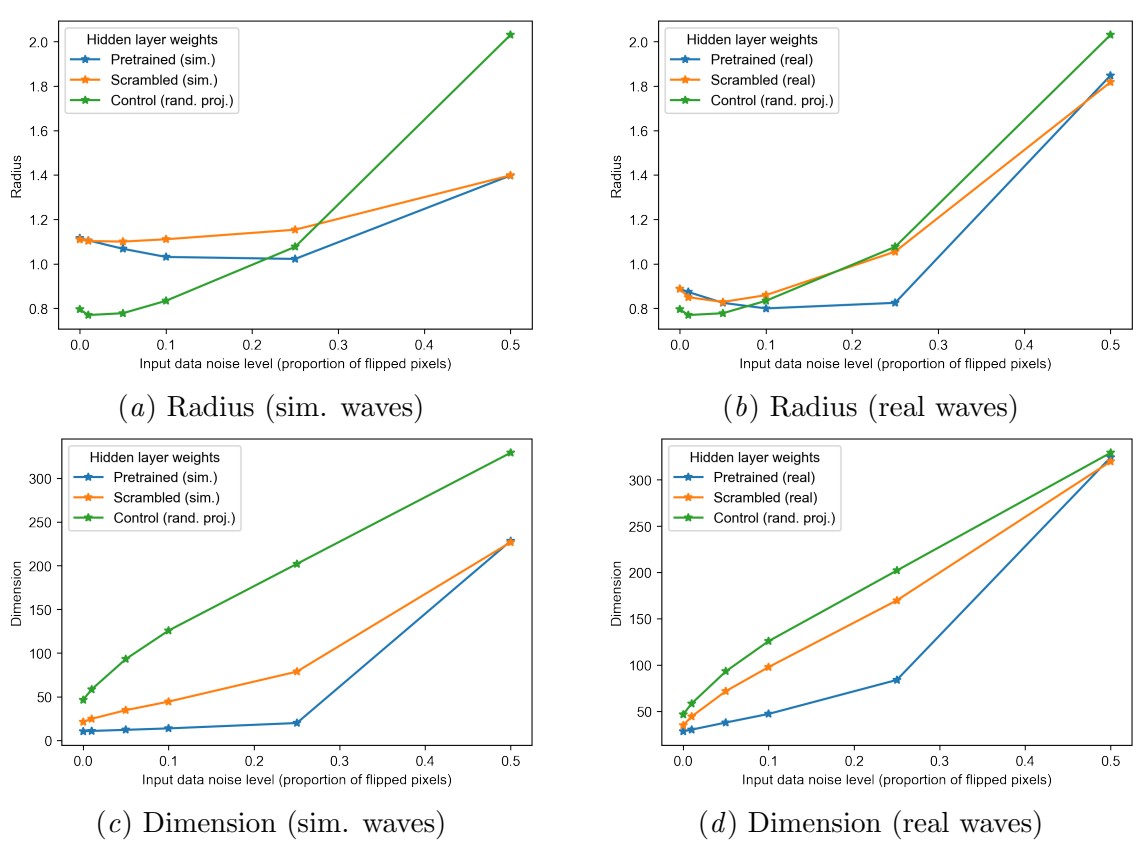

Figure 2: Manifold geometry of hidden layer representations averaged across digit classes.

2001). All networks considered have the same width and depth (Appendix B). We find that classification accuracy is higher for pre-trained networks relative to their scrambled counterparts. Pre-trained networks also maintain higher classification performance given noisy input data (Fig. 1(a)).

We next analyze three manifold properties of the networks' internal representations: capacity, defined as the maximum number of manifolds that can be linearly separated using random binary labels divided by the dimension of the representation; manifold dimension, the spread of anchor points (which define the optimal separating hyperplane between two perceptual manifolds) along the manifold axes; and manifold radius, the variance of anchor points normalized by the average distance between manifold centers. The lower the dimension and radius of the object manifolds, the more linearly separable they are, yielding higher manifold capacity (Fig. 1(b)) and therefore higher accuracy (Fig. 1(a)) (Chung et al., 2018). We show that the internal representations of pre-trained networks generally exhibit higher capacity, lower dimension, and lower radius relative to those of their scrambled counterparts, particularly in the presence of moderate noise. (Fig. 2). Notably, the pre-trained networks exhibit consistently lower dimension than the control (random projection) network (Fig. 2(c), 2(d)), but lower radius only in the presence of considerable noise (Fig. 2(a), 2(b)). Moreover, the dimension of the networks pre-trained on simulated waves tends to be lower than that of networks pre-trained on real waves, while the real wave networks tend to exhibit lower radius.

## 3. Discussion

Our results show that pre-training with both simulated and real retinal waves yields receptive fields with spatial structure favorable for separation and classification of object manifolds. Because we are comparing to scrambled networks, whose receptive fields share the same pixel distribution as the pre-trained networks but lack their spatial structure, these results suggest that learning the spatial structure in retinal wave patterns, as opposed to just the overall distribution, is relevant for object recognition. We also find that networks trained on simulated waves exhibit higher accuracy and manifold capacity compared to networks trained on real retinal waves (Fig. 1). We suspect this is because the real data contains considerably more noise and is less local in its spatial structure (Fig. C), which may lower performance on the relatively simple binarized MNIST dataset.

Interestingly, it appears simulated wave and real wave networks have different effects on the manifold geometry. Simulated wave networks have a greater effect on reducing the manifold dimension, while real wave networks have a greater effect on reducing the manifold radius (Fig. 2). The disparity in dimension reduction may be due to the local structure of the simulated wave receptive fields, which could act as feature detectors for low dimensional structures induced by correlated nearby pixels. The reasons for disparity in radius reduction do not appear as straightforward, considering that the random projection network tend to exhibit lower radii. One factor could be that the high magnitude of the synaptic strengths induced by the slower, localized simulated waves increases the effective sizes of the data manifolds and thus their radii. The real retinal waves propagate less frequently and diffusely, so the synaptic strengthening during the Hebbian learning phase occurs at a lower rate. This explanation is consistent with the fact that the random projections have unit magnitude

Extended Abstract Track

and generally lower radii, though it is not clear why their radii are higher in noisier regimes. Normalizing the receptive fields in the pre-trained networks to control for the effect of synaptic strength on radius may elucidate these questions

Finally, we highlight that the task (as well as the dynamics of the simulated retinal waves) were chosen as a simple benchmark to compare with previous similar work (Raghavan and Thomson, 2019) and as a proof of concept, limiting the scope of our findings. A more ethologically relevant task, like classifying natural images (without binarization of the pre-training and training data, as is done here), would be a more direct examination of the role of retinal waves in biological development and for this reason may be more amenable for networks pre-trained on real data. Another direction for future work is to examine the effect of network architecture in this regime, in particular by introducing layers that more accurately mimic the structure and dynamics of higher visual areas like LGN and V1.

## 4. Methods

The model architecture considered is a three layer feed-forward network consisting of a 3200-unit input layer of retinal ganglion cells randomly placed on a square grid, a 400-unit hidden layer with ReLU activations, and a 1 unit classifier/output layer (Appendix B). Retinal waves are propagated across the input layer to train the weights (receptive fields) to the first hidden layer units by a winner-take-all Hebbian learning rule as in (Raghavan and Thomson, 2019). The MNIST data is pre-processed by binarizing the pixel values and projecting them onto the input layer units. Noisy MNIST datasets are generated by bit-flipping a randomly selected proportion of the pixels.

After pre-training, the network is trained by freezing the input and hidden layer weights as is, doing a forward pass of the projected MNIST data through the second hidden layer, and training the classifier layer, a support vector machine, on the 400-dimensional hidden layer output representations of the data. Once trained, models are then evaluated on test data at the same given noise level as the training set. For each noise level, the training set consists of 60,000 samples and the test set consists of 10,000 samples.

Manifold analysis is done for the second hidden layer output representations of each digit class and averaged across the classes according to the procedure described in Chung et al. (2018). The code for pre-processing, pre-training, MNIST training, and manifold analysis is publicly available in Jupyter notebooks at https://github.com/ligeralde/wave_manifold.

## Acknowledgments

We thank Michael R. DeWeese and the members of the Redwood Center for Theoretical Neuroscience for many useful discussions.

## References

David A. Arroyo and Marla B. Feller. Spatiotemporal Features of Retinal Waves Instruct the Wiring of the Visual Circuitry. *Frontiers in Neural Circuits*, 10:54, July 2016. ISSN 1662-5110. doi: 10.3389/fncir.2016.00054. URL https://www.ncbi.nlm.nih.gov/pmc/articles/PMC4960261/.

# Extended Abstract Track

Ella Bingham and Heikki Mannila. Random projection in dimensionality reduction: applications to image and text data. In *KDD '01*, 2001.

Jianhua Cang, René C. Rentería, Megumi Kaneko, Xiaorong Liu, David R. Copenhagen, and Michael P. Stryker. Development of precise maps in visual cortex requires patterned spontaneous activity in the retina. *Neuron*, 48(5):797–809, December 2005. ISSN 0896-6273. doi: 10.1016/j.neuron.2005.09.015.

Anand R. Chandrasekaran, Daniel T. Plas, Ernesto Gonzalez, and Michael C. Crair. Evidence for an instructive role of retinal activity in retinotopic map refinement in the superior colliculus of the mouse. *The Journal of Neuroscience: The Official Journal of the Society for Neuroscience*, 25(29):6929–6938, July 2005. ISSN 1529-2401. doi: 10.1523/JNEUROSCI.1470-05.2005.

SueYeon Chung and L. F. Abbott. Neural population geometry: An approach for understanding biological and artificial neural networks. *Current Opinion in Neurobiology*, 70:137–144, October 2021. ISSN 09594388. doi: 10.1016/j.conb.2021.10.010. URL http://arxiv.org/abs/2104.07059. arXiv: 2104.07059.

SueYeon Chung, Daniel D. Lee, and Haim Sompolinsky. Classification and Geometry of General Perceptual Manifolds. *Physical Review X*, 8(3):031003, July 2018. ISSN 2160-3308. doi: 10.1103/PhysRevX.8.031003. URL https://link.aps.org/doi/10.1103/PhysRevX.8.031003.

James J. DiCarlo, Davide Zoccolan, and Nicole C. Rust. How Does the Brain Solve Visual Object Recognition? *Neuron*, 73(3):415–434, February 2012. ISSN 0896-6273. doi: 10.1016/j.neuron.2012.01.010. URL https://www.sciencedirect.com/science/article/pii/S089662731200092X.

J. Sebastian Espinosa and Michael P. Stryker. Development and Plasticity of the Primary Visual Cortex. *Neuron*, 75(2):230–249, July 2012. ISSN 08966273. doi: 10.1016/j.neuron.2012.06.009. URL https://linkinghub.elsevier.com/retrieve/pii/S0896627312005697.

M. B. Feller and D. Kerschensteiner. Chapter 16 - Retinal waves and their role in visual system development. In John Rubenstein, Pasko Rakic, Bin Chen, Kenneth Y. Kwan, Hollis T. Cline, and Jessica Cardin, editors, *Synapse Development and Maturation*, pages 367–382. Academic Press, January 2020. ISBN 978-0-12-823672-7. doi: 10.1016/B978-0-12-823672-7.00016-8. URL https://www.sciencedirect.com/science/article/pii/B9780128236727000168.

Andrew D. Huberman, Colenso M. Speer, and Barbara Chapman. Spontaneous Retinal Activity Mediates Development of Ocular Dominance Columns and Binocular Receptive Fields in V1. *Neuron*, 52(2):247–254, October 2006. ISSN 0896-6273. doi: 10.1016/j.neuron.2006.07.028. URL https://www.cell.com/neuron/abstract/S0896-6273(06)00625-8. Publisher: Elsevier.

Jonathan J. Hunt, Michael Ibbotson, and Geoffrey J. Goodhill. Sparse Coding on the Spot: Spontaneous Retinal Waves Suffice for Orientation Selectivity. *Neural Computation*, 24

(9):2422–2433, September 2012. ISSN 0899-7667. doi: 10.1162/NECO_a_00333. URL https://doi.org/10.1162/NECO_a_00333.

Jeffrey Markowitz, Yongqiang Cao, and Stephen Grossberg. From Retinal Waves to Activity-Dependent Retinogeniculate Map Development. *PLOS ONE*, 7(2):e31553, February 2012. ISSN 1932-6203. doi: 10.1371/journal.pone.0031553. URL https://journals.plos.org/plosone/article?id=10.1371/journal.pone.0031553. Publisher: Public Library of Science.

Guruprasad Raghavan and Matt Thomson. Neural networks grown and self-organized by noise, June 2019. URL http://arxiv.org/abs/1906.01039. Number: arXiv:1906.01039 arXiv:1906.01039 [nlin, q-bio].

## Appendix A. In vitro retinal wave data preparation

Mice aged postnatal day 8-11 (P8-P11) were deeply anesthetized with isoflurane inhalation and euthanized by decapitation. Eyes were immediately enucleated and retinas were dissected at room temperature under infrared illumination in oxygenated (95% $O_2$/5% $CO_2$) artificial cerebrospinal fluid (ACSF) (in mM, 119 NaCl, 2.5 KCl, 1.3 $MgCl_2$, 1 $K_2HPO_4$, 26.2 $NaHCO_3$, 11 D-glucose, and 2.5 $CaCl_2$). Cuts along the chloride fissure were made prior to isolating the retina from the retinal pigmented epithelium. These cuts were made to precisely orient retinas. Isolated retinas were mounted whole on filter paper with the photoreceptor layer side down, and transferred in a recording chamber of a microscope for subsequent imaging. The whole-mount retinas were continuously perfused (3 mL/min) with oxygenated ACSF media at 32-34°C for the duration of the experiment. The retina from the other eye was kept in the dark at room temperature in ACSF media bubbled with 95% $O_2$, 5% $CO_2$ until use (maximum 8 h).

Retinas of C57B6 mice used for two-photon calcium imaging were mounted over white filter paper (Watman) and bolus loaded with either the green calcium dye Calbrtye 520 AM or the red calcium dye Calbryte 590 AM. Two-photon fluorescence measurements were obtained with a modified movable objective microscope (MOM) (Sutter instruments, Novator, CA) and made using a Nikon 16×, 0.80 NA, N16×LWD-PF objective (Nikon, Tokyo, Japan). Two-photon excitation of calcium dyes was evoked with an ultrafast pulsed laser (Chameleon Ultra II; Coherent, Santa Clara, CA) tuned to 920 nm for green dyes and GFP or 1040 nm for red dyes. The microscope system was controlled by ScanImage software (www.scanimage.org). Retinal waves were imaged with the field of view set to 850um×850um and the scan parameters were [pixels/line×lines/frame (frame rate in Hz)]: [128×128 (5.92 Hz)], at 1 ms/line. The same field of view was imaged until at least 50 retinal waves had propagated through it for a total of 18,000 images.

To ensure a sufficient number of pre-training steps, the original set of 18,000 images was augmented by appending 3 rotations of the images to the data (90°, 180°, and 270°), for a total of 72,000 images. The images were then mean-pooled with a block size of 4, and the pixel intensities were projected onto the retina as binary values as described above in Section 4.

## Appendix B. Model architecture

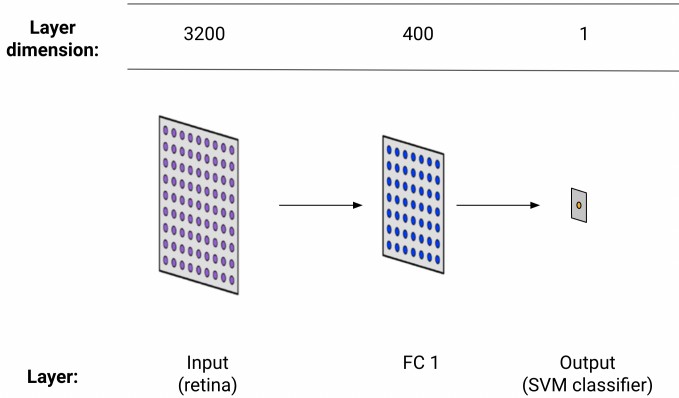

Figure 3: 3-layer feed-forward network with ReLU activations after the hidden layer.

The model architecture in Fig. 3 is the same for all networks considered, with the only difference being the network parameters, in particular the feedforward weights between the input layer and FC 1 (the fully connected hidden layer) and the hyperplane learned by the SVM.

## Appendix C. Receptive fields of pre-trained and shuffled networks

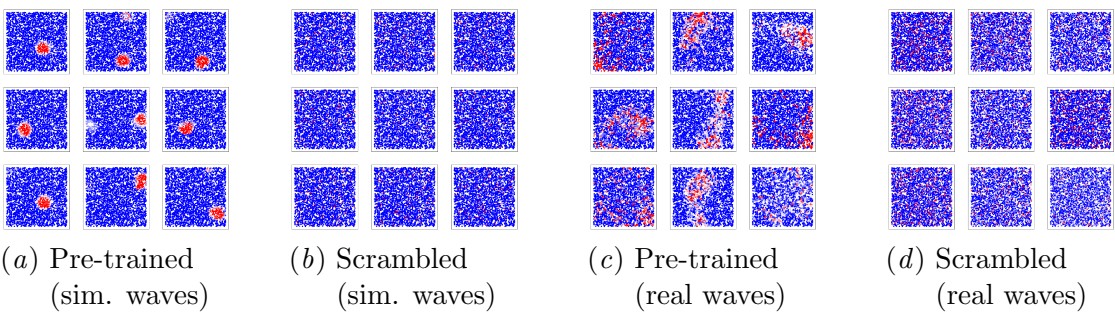

($a$) Pre-trained    ($b$) Scrambled    ($c$) Pre-trained    ($d$) Scrambled
(sim. waves)    (sim. waves)    (real waves)    (real waves)

Figure 4: Receptive fields have spatial structure that resembles that of the retinal waves used in pre-training. Each scrambled receptive field is a random permutation of the pixels in the corresponding pre-trained receptive field.

