# OpenReview forum: "Geometry reveals an instructive role of retinal waves as biologically plausible pre-training signals"
_NeurIPS.cc/2022/Workshop/NeurReps — NeurReps 2022 Poster_

### Official Review · Reviewer_6S4n · 2022-10-14
**Conceptually lacking approach to understand the computational role of retinal waves in the developing brain.**

**Confidence:** 3
**Soundness:** 2
**Presentation:** 3
**Contribution:** 2
**Overall Rating:** 5

**Summary:**

This work studies the computational role of retinal waves (i.e. correlated activity patterns in the developing retina). The authors trained —with a Hebbian learning rule— the first layer of a three-layer neural network on simulated / recorded input retina wave patterns. They then trained the last layer on MNIST and measure its classification accuracy along with other object manifold geometry metrics. They found that training the first layer on retinal waves  leads to higher classification accuracy than the random weights control.

**Questions:**

- Does initializing the network with the weights resulting from pre-training on retinal waves (compared to random) lead to better object separability (or any other benefit) after training the whole network on visual inputs with either supervised or unsupervised objectives? Aren’t natural images enough to induce retinal spatiotemporal receptive fields (e.g. ON-OFF center surround RFs) from scratch?
- What is the effect of dimensionality (width of the network) when comparing random vs pretrained?
- What is the effect of depth when comparing random vs. pretrained? Presumably classes will be more separable if you add nonlinearities (even when the first layer has random weights).

**Limitations:**

The authors could also discuss the limitations on running this analysis on a simple binarized set of stimuli (MNSIT).

**Recommended Decision:**

2: Borderline

**Relevance:**

2: Limited relevance

**Strengths And Weaknesses:**

Although the work is certainly going in an interesting direction, I find that the current analysis is conceptually unsound to solve the central question.

If the authors want to test wether unsupervised pretraining on retinal waves facilitates object classification in higher areas of the ventral stream, why not consider the effect of such pretraining after visual experience on natural images is made available? In the end, what is of most ethological relevance to the animal is to discriminate objects in the world during actual visual experience. If the end result (e.g. structured RFs) can be similarly achieved with unsupervised rules applied to natural image stimuli once it is available, what’s the benefit of these retina patterns?

Strengths:

- The role of wave patterns in the developing retina is of great interest in vision neuroscience
- The authors use real recordings of retinal waves to support their claims. These recordings are of great value!
- The authors also make comparisons with models of simulated retina waves.

Weaknesses

- The accuracy on MNIST was too low for a 3 layer network (Fig 1). A single linear SVM trained on pixels achieves 92% accuracy already ([https://dmkothari.github.io/Machine-Learning-Projects/SVM_with_MNIST.html](https://dmkothari.github.io/Machine-Learning-Projects/SVM_with_MNIST.html)) This weakens the validity of the claims.
- Object separability is measured on a single dataset with little resemblance to natural images (MNIST) that is far from real visual experience.
- There was little motivation (biological or otherwise) for only pretraining the first layer on retinal waves (and not deeper ones). There was also no correspondence between network layers and areas in the brain that the model was attempting to match.
- There were no comparisons between initializing the network with pretrained weights before and after visual experience.
- The claims depend strongly on network choices (e.g. width and depth)
- Link to code disservice double blind review.

**Submission Track:**

Extended Abstract (4 Page)

---

> ### Author Response · Authors · 2022-11-02
> **Clarification of experimental choices and pre-training vs. training**
>
> Thank you for your review and your helpful feedback! Copied questions/concerns below with my responses as revisions and answers.
>
> Q: The accuracy on MNIST was too low for a 3 layer network (Fig 1). A single linear SVM trained on pixels achieves 92% accuracy already (https://dmkothari.github.io/Machine-Learning-Projects/SVM_with_MNIST.html) This weakens the validity of the claims.
>
> A: Our accuracy now more closely reflects the 92% accuracy (around 90% for the noiseless case in all networks). We found removing the untrained layer achieves better accuracy in all regimes. We also note that the MNIST data we use is not the original MNIST, but a binarized projection onto a model 'retina', so the accuracy won't necessarily reflect SVM on the raw data.
>
> Q: Object separability is measured on a single dataset with little resemblance to natural images (MNIST) that is far from real visual experience.
>
> A: We mainly used MNIST as it is a simple task and benchmark for a model used in a previous study that we were trying to first replicate (Raghavan & Thomson, referenced in abstract). It is true, MNIST is not the most natural task! We address this in the discussion now more clearly, that we used MNIST as a proof of concept but want to extend to more ethologically relevant tasks like natural images.
>
> Q:  There was little motivation (biological or otherwise) for only pretraining the first layer on retinal waves (and not deeper ones).
>
> A: See above (the untrained hidden layer was removed)
>
> Q: There was also no correspondence between network layers and areas in the brain that the model was attempting to match.
>
> A: Fair concern. We used this architecture because it is simple, feedforward, and easy to analyze, and more importantly to closely mimic the previous work we were trying to first recreate. But yes, we want to more accurately model Retina/LGN. For instance, using dynamics and numbers of cells in the hidden layer that correspond to those in Retina/LGN.
>
> Q: There were no comparisons between initializing the network with pretrained weights before and after visual experience.
>
> A: The goal of this study is to mimic the course of development. The pre-training phase mimics the experience-independent phase of development. While it is true that visual experience is sufficient to learn the RFs in RGC and higher visual areas, it is not necessary; the connections that enable a lot of visual functions are well established prior to the visual experience, and presumably, the ones responsible for object recognition are, too. So we wanted to make a connection between the pre-training phase and enabling object recognition, since this is presumably what's happening before the onset of vision. To answer the question, for this reason we did not look at pre-trained weights after the visual experience. The visual experience, to be clear, has no bearing on the receptive fields in this work since they are frozen once the images are presented to the network - the images only train the svm (output). So we are really studying the representation induced by the pre-training. We do have a network without pre-training and only visual experience (see abstract) as a control, but pre-training after the training would not reflect the goal of this work; maybe i'm misunderstanding, but I'm not clear on how we would train the SVM and then go back to pre-training the first layer/what this would show.
>
> Q: The claims depend strongly on network choices (e.g. width and depth)
>
> A: They do - we address this now as a limitation. We only wanted to look at fairly simple networks with one set of receptive fields, but in future work we want to vary the width and layers (i.e., size and number of higher visual areas).
>
>
> Q: Does initializing the network with the weights resulting from pre-training on retinal waves (compared to random) lead to better object separability (or any other benefit) after training the whole network on visual inputs with either supervised or unsupervised objectives? Aren’t natural images enough to induce retinal spatiotemporal receptive fields (e.g. ON-OFF center surround RFs) from scratch?
>
> A: See above.
>
> Q:What is the effect of dimensionality (width of the network) when comparing random vs pretrained? What is the effect of depth when comparing random vs. pretrained? Presumably classes will be more separable if you add nonlinearities (even when the first layer has random weights).
>
> A: See above - and also, I neglected to mention this in the first round, but I've now included the fact that we used nonlinearties after the hidden layer.

---

### Official Review · Reviewer_oLAi · 2022-10-15
**Fascinating research question, thorough evaluation, concerns in the soundness of the experimental setting.**

**Confidence:** 4
**Soundness:** 2
**Presentation:** 2
**Contribution:** 3
**Overall Rating:** 5

**Summary:**

This work analyzes the usage of retinal waves as a pre-training signal for machine learning classifiers. This is achieved by comparing different initializations of the model: one where the retina waves are used as pre-trained features and a control one where they are randomly permuted before being used. The evaluation is performed both via a task-driven metric (MNIST classification) and a geometry-driven one (manifold capacity). The results suggest that retinal waves are indeed informative features to be used in image classification tasks.

**Questions:**

Have the authors tried to apply the SVM directly on the input features instead of adding the two hidden layers before it? I’m asking this because I expect the same gap between the clean and the shuffled versions, but lower scores overall.

**Limitations:**

I did not find any additional limitations to include here.

In any case, I’m more than open to changing my ratings and recommended decision if the main concerns/weaknesses are addressed.

**Recommended Decision:**

2: Borderline

**Relevance:**

4: Highly relevant

**Strengths And Weaknesses:**

My expertise is not in biology, but concerning that part, I did not have problems following the work. Therefore, I consider this a strength of the paper and thank the authors for keeping it more than accessible for researchers with different backgrounds. Not only is it well-motivated in the introduction, but it also is fascinating to see machine learning being used as a proxy to explore hypothesis biology-related: in this case, whether retina waves retain enough information to be used as features to discern between samples in a simple image classification task.

My main concerns are in the experimental setup (clearly, excluding the data collection, which is well detailed in the appendix). I had some issues getting a clear picture of the model structure both in the first part of section 2 (Results) and in section 4 (Method). I think that portion could benefit from some restructuring; maybe a figure could be of help. However, the main issue was not understanding the number and nature of each layer (which is clearly illustrated) but the model as a whole and its design choices. Especially in the experimental setting, many choices are left unjustified, which is, in my opinion, weakening the final discussion and conclusions. For example:

- Why choose first to embed the retina features via Hebbian learning instead of using them directly as input? That could add noise to the procedure instead of using the raw data.
- Why is the second hidden layer (the one mapping from 400 to 100 dimensions) not trained? It has been shown in the literature that simple dense layers randomly initialized can still reach good classification performances, but why the need to add one here instead of just removing it if it can't be trained?
- How are the simulated samples obtained? I’m curious about this because I was expecting the real and simulated versions to have a similar distribution. However, in Figure 3, they are shown as being quite dissimilar, especially when comparing the two scrambled versions.

To be clear: if I understood correctly, the experimental setting is fairly solid. The problem is that I found it a bit convoluted to comprehend completely, mainly because there are some non-standard design choices left unmotivated, which negatively impacts the soundness of the results.

**Submission Track:**

Extended Abstract (4 Page)

---

> ### Author Response · Authors · 2022-11-02
> **Addressing some model/experiment concerns**
>
> Thank you for your review and your helpful feedback! Copied questions/concerns below with my responses as revisions and answers.
>
> Q: My main concerns are in the experimental setup (clearly, excluding the data collection, which is well detailed in the appendix). I had some issues getting a clear picture of the model structure both in the first part of section 2 (Results) and in section 4 (Method). I think that portion could benefit from some restructuring; maybe a figure could be of help.
>
> A: The answer to most of these questions is that we wanted a model that could be a benchmark with a previous work (Raghavan & Thomson, cited in abstract) and compare our results to. So we started by recreating their result, and tweaking our model based on our findings. Figure of the model is now in the abstract.
>
> Q: Why choose first to embed the retina features via Hebbian learning instead of using them directly as input? That could add noise to the procedure instead of using the raw data.
>
> A: This was an artifact of the study we were trying to recreate. In the future, we want to use these directly as input and without binarizing.
>
> Q: Why is the second hidden layer (the one mapping from 400 to 100 dimensions) not trained? It has been shown in the literature that simple dense layers randomly initialized can still reach good classification performances, but why the need to add one here instead of just removing it if it can't be trained?
>
> A: We already found that removing the untrained layer (an artifact of Raghavan & Thomson) is actually better for network performance. The abstract is revised to reflect this change.
>
> Q: How are the simulated samples obtained? I’m curious about this because I was expecting the real and simulated versions to have a similar distribution. However, in Figure 3, they are shown as being quite dissimilar, especially when comparing the two scrambled versions.
>
> A: The simulated samples are based on the dynamics of the Raghavan & Thomson simulated waves. Certainly there are many models of retinal waves one could use. This model provided a contrast with the real waves (more local features), but more importantly, a benchmark/sanity check to compare with their work. We're exploring different kinds of simulated waves in the future that we can tweak to reflect different image statistics (power spectra, for instance) and may be optimal for different tasks (e.g., natural images).

---

### Official Review · Reviewer_WG7u · 2022-10-16
**Interesting study showing how retinal waves combined with a hebbian learning rule can be leveraged to improve manifold separability.**

**Confidence:** 4
**Soundness:** 4
**Presentation:** 4
**Contribution:** 3
**Overall Rating:** 7

**Summary:**

This abstract shows that pre-training a classifier with a biologically plausible Hebbian learning rule on "retinal wave"-like patterns improves the separability of the network’s internal representations. Using a sophisticated geometrical framework to analyze manifolds, they show that the manifolds' reduced dimension as opposed to their radius is the primary driver of the increase in capacity of the network.




**Questions:**

- It is an interesting finding that the manifolds' reduced dimension as opposed to their radius is the primary driver of the increase in capacity of the network, but how to interpret this finding? Could it be for example that, thanks to their localized receptive fields, neurons pool from nearby correlated pixels, effectively reducing the dimensionality of the manifold?



**Limitations:**

Limitations are adequately addressed.

**Recommended Decision:**

3: Accept

**Relevance:**

4: Highly relevant

**Strengths And Weaknesses:**

Significance: The theoretical findings are interesting for our understanding of the role of retinal waves in neuroscience. For machine learning, the practical implications of retinal-wave learning is not demonstrated in this abstract, because of the simplicity of the task (MNIST) and architecture (one hidden layer, no backprop) studied.

Originality: I haven't seen such a precise geometrical analysis on the benefits of retinal-wave pre-training for manifold separability.

Quality/ Clarity: The abstract is clearly written and the claims well supported by the evidence.


**Submission Track:**

Extended Abstract (4 Page)

---

> ### Author Response · Authors · 2022-11-02
> **Interpretation of radius versus dimension**
>
> Thank you for your review and your question. I incorporated this explanation into the abstract. TLDR: It may be an artifact of synaptic strength.
>
> Interestingly, it appears simulated wave and real wave networks have different effects on the manifold geometry. Simulated wave networks have a greater effect on reducing the manifold dimension, while real wave networks have a greater effect on reducing the manifold radius. I think your comment is relevant here: disparity in dimension reduction may be due to the local structure of the simulated wave receptive fields, which could act as feature detectors for low dimensional structures induced by correlated nearby pixels. The reasons for disparity in radius reduction do not appear as straightforward, considering that the random projection network tends to exhibit lower radii. One factor could be that the high magnitude of the synaptic strengths induced by the slower, localized simulated waves increases the effective sizes of the data manifolds and thus their radii. The real retinal waves propagate less frequently and diffusely, so the synaptic strengthening during the Hebbian learning phase occurs at a lower rate. This explanation is consistent with the fact that the random projections have unit magnitude and generally lower radii, though it is not clear why their radii are higher in noisier regimes. Normalizing the receptive fields in the pre-trained networks to control for the effect of synaptic strength on radius may elucidate these questions.

---

### Decision · Program_Chairs · 2022-10-21

Accept (Poster)